# Machine Learning Approach for Determining the Formation of β-Lactam Antibiotic Complexes with Cyclodextrins Using Multispectral Analysis

**DOI:** 10.3390/molecules24040743

**Published:** 2019-02-19

**Authors:** Mikołaj Mizera, Kornelia Lewandowska, Andrzej Miklaszewski, Judyta Cielecka-Piontek

**Affiliations:** 1Department of Pharmacognosy, Faculty of Pharmacy, Poznań University of Medical Sciences, Święcickiego 4, 60-781 Poznań, Poland; mikolajmizera@gmail.com; 2Institute of Molecular Physics, Polish Academy of Science, ul. Smoluchowskiego 17, 60-179 Poznań, Poland; kornelia.lewandowska@ifmpan.poznan.pl; 3Institute of Materials Science and Engineering, Poznan University of Technology, Pl. M.Sklodowskiej-Curie 5, 60-965 Poznań, Poland; andrzej.miklaszewski@put.poznan.pl

**Keywords:** cyclodextrin, β-lactam antibiotic, multispectral analysis, machine learning, molecular modeling

## Abstract

The problem of determining the formation of complexes of β-lactam antibiotics with cyclodextrins (CDs) and the interactions involved in this process were addressed by machine learning on multispectral images. Complexes of β-lactam antibiotics, including cefuroxime axetil, cefetamet pivoxil, and pivampicillin, as well as CDs, including αCD, βCD, γCD, hydroxypropyl-αCD, methyl-βCD, hydroxypropyl-βCD, and hydroxypropyl-γCD, were prepared in all combinations. Thermograms confirming the formation of cyclodextrin complexes were obtained using differential scanning calorimetry. Transmission Fourier-transform infrared (tFTIR) and complementary attenuated total reflectance FTIR (ATR) coupled with machine learning were techniques chosen as a nondestructive alternative. The machine learning algorithm was used to determine the formation of complexes in samples using solely their tFTIR and ATR spectra at the prediction stage. Parameterized method 7 (PM7) was used to support the analysis by molecular modeling of the complexes. The model developed through machine learning properly distinguished samples with formed complexes form noncomplexed samples with a cross-validation accuracy of 90.4%. Analysis of the contribution of spectral bands to the model indicated interactions of ester groups of β-lactam antibiotics with CDs, as well as some interactions of cephem ring in cefetamet pivoxil and penam moiety in pivampicillin. Molecular modeling with PM7 helped to explain experimental results and allowed to propose possible binding modes.

## 1. Introduction

The active pharmaceutical ingredients (APIs) systems with macromolecules such as cyclodextrins (CDs) are a promising way of enhancing physicochemical properties of well-established drugs. The analysis of such systems is the major challenge in the field of quality control due to a problem with the possible different stoichiometric ratio between reagents. An innovative approach for the analysis of the cyclodextrin systems may be based on theoretical methods involving machine learning and molecular modeling. The in silico techniques are particularly valuable when the included substances are APIs, which exhibit many beneficial physicochemical changes when they are in the cavity of a cyclodextrin carrier. Therefore, as a model, APIs in this work as esters of selected derivatives of β-lactam analogs (prodrugs) was proposed. The benefits of combining prodrugs of β-lactam analogs with cyclodextrins include the possibility of modifying their solubility, increasing absorption, masking the specific taste, and even increasing bactericidal effectiveness [1,2].

Prodrugs are poorly or non-active chemical compounds which are biotransformed in vivo to their active form during any stage of absorption, distribution, and metabolism. As defined, the prodrug can be chemical bioprecursor undergoing transformation to an active drug or carrier-linked drug, where the active form is attached to a chemical group such as an ester, amide, or carbamate. Also, another active compound performing a pharmaceutical action can be linked, forming a mutual prodrug which simultaneously releases two active molecules after transformation [3]. Development of prodrugs is a way of addressing issues met during the early stages of API development such as low solubility [4], chemical instability [5], and bad taste [6]; also, changes of pharmacokinetic properties may be achieved, including permeation through the gastrointestinal tract [7], blood–brain penetration [8], modification of therapeutic activity time [9], and pharmacodynamic properties, including toxicity reduction of some antitumor drugs [10] and improvement of therapeutic index [11]. Development of prodrugs by linking with lipophilic moiety may have a negative impact on solubility compared to the parent molecule. As a result, lowered bioavailability creates secondary issues which should be taken into consideration when formulating a drug [12]. Low solubility can be addressed in several ways, including salt formation, development of cocrystals [13], amorphous solid dispersions [14], nanocrystals [15], cryogenic co-grinding [16], and forming complexes with cyclodextrins [17]. Cyclodextrins are useful multifunctional excipients used for the modification of important properties of the parent molecule such as solubility, chemical and physical stability, permeability, and delayed release in the case of pharmaceutical dosage forms [18]. Cyclodextrins include natural cyclodextrins comprising six to eight glucose subunits and modified cyclodextrins substituted with others, including hydroxypropyl, methyl, and sulfobutylether moieties. The specific distribution of functional groups on both sides of linked glucose subunits creates a toroidal shape with a polar surface outside and the lipophilic cavity inside the macromolecule. A number of glucose subunits, as well as chemical modifications, influence the physicochemical properties of the polymer, causing differences in the size of the cavity, polarity, and solubility, thus facilitating the ability to form complexes with specific kinds of guest molecules. Preparation techniques of inclusion complexes can be divided into two main groups: methods carried out in a solid state such as kneading and grinding, and methods performed in media. Grinding of powder API and cyclodextrins were used in the preparation of the β-cyclodextrin complex with acetaminophen, warfarin, indometacin, diazepam, and hydrocortisone acetate [19]. However, the coprecipitation method requires saturated solutions of guest molecules and saturated water solutions of cyclodextrin. The method was applied to prepare, among others, cyclodextrin complexes of silymarin and ketoconazole [20,21].

A wide range of analysis techniques was used for the identification of cyclodextrin complexes. The common techniques are FTIR and/or Raman spectroscopy, differential scanning calorimetry (DSC) and thermogravimetric thermal methods, X-ray crystallography, X-ray powder diffraction, and microscopic imaging techniques [22,23,24]. The DSC technique is the most preferable due to reliability and relative ease of analysis. However, the DSC technique is not specific enough to indicate domains involved in the interaction in guest–host inclusion systems. Moreover, the major drawback of thermal techniques is their destructive nature. Spectral methods, on the other hand, offer more complex measurement results, allowing a deeper insight into the kinds of interactions between API and cyclodextrin. Therefore, they can be used as an alternative method to DSC. Significant benefits of their application include their nondestructive nature and the ability for observations of close-range interactions, making it valuable for analysis of cyclodextrin carriers in inclusion complexes with APIs. However, an interpretation of the multidimensional data coming from the spectral analysis is complex, especially when identification of API domains interacting with cyclodextrins is required. Due to the high dimensionality of spectral data, the analysis may be an objective for a computer-aided approach. Machine learning techniques are applied in the areas where due to complexity or a large amount of data, analysis by a human may be impossible in the required time. In some areas of analytical sciences, machine learning algorithms helped to significantly improve analysis by speeding up the process or allowing analytical techniques to change to one which is more ecologically, cost-, and time-efficient. In the area of electrochemistry, machine learning techniques allowed scientists to quantify electrolyte salts based on FTIR measurements, meeting the accuracy level of gas chromatography and inductively coupled plasma optical emission spectrometry [25]. In a more general approach, the classifier was proposed to determine the presence of molecular groups in FTIR spectra, resulting in less error-prone techniques comparing to human inspection [26]. Detection of glioma stem cell differentiation in hyperspectral data was handled with principal component analysis (PCA) logistic regression, decision tree, random forest, k-nearest neighbors, and hierarchical cluster analysis, resulting in a high accuracy of analysis [27]. Wavelet-based techniques were applied for feature extraction from FTIR data for classification of plant species [28]. The PCA and support vector machine were used for the discrimination of grape seeds basing on their hyperspectral images [29].

Development of computer-aided analysis algorithms of cyclodextrin complexes has not been reported to the best of our knowledge. The desired method should be able to detect the presence of the complex in a given sample when provided with the spectrum of a complex sample and the spectra of pure ingredients. Assuming a proper spectral dataset of complexed and pure API, machine learning is a relevant technique that is able to learn dependencies directly from spectral data. 

As model APIs, the following prodrug β-lactam analogs were proposed: cefuroxime axetil (CA), cefetamet pivoxil (CT), and pivampicillin (PA). The prodrugs compared to acidic forms of selected β-lactam analogs are characterized by better bioavailability, more acceptable taste, and greater chemical stability. As a consequence of changing the acidic nature of hydrophilic β-lactam analogs to lipophilic prodrugs, the solubility decreased significantly. Considering the solubility issue of prodrugs, the use of solubilizers as drug carriers in their formulation such as cyclodextrins is fully justified. In this paper, α-, β-, and γ-cyclodextrins (αCD, βCD, γCD, respectively) were used along with their synthetic analogs, including hydroxypropyl derivatives of all used natural CDs (HPαCD, HPβCD, HPγCD) as well as methyl-βCD (mβCD) as carriers. 

The aim of this work was to prepare complexes of prodrug β-lactam analogs and CDs and to confirm complex formation based on analysis of spectral and thermal measurements, followed by a development of a machine-learning-based model for the determination of complex formation in samples, based solely on ATR and tFTIR spectra with simultaneous identification of domains involved in the creation of the complex.

## 2. Results and Discussion

### 2.1. Determination of Complexes Formation

DSC was used as the reference method to determine the formation of prodrug β-lactam analogs–CDs systems. The DSC thermograms (Appendix A) of investigated systems were analyzed, and the results were presented in Table 1. For all prodrug β-lactam analogs–CD systems except CA–HPαCD, the complex formation was confirmed. In CA–CDs samples, the main peak of interest was the one corresponding to glass transition of the API around Tg = 86 °C. Analysis of thermal curves revealed that all samples except the one with HPαCD expose changes in complexed sample thermograms when compared to the physical mixture. Complexation occurred also in all samples of CT complexes, where changes around Tg = 163 °C were considered and in all samples of PA where main changes were noted around the glass transition point Tg = 138 °C.

The second part of the experimental studies was the spectral analysis of the prodrug β-lactam analogs–CDs complexes obtained in relation to the changes of the positions of changes of the bands and vibration intensity observed for the APIs and cyclodextrins themselves. Using the theoretical approach, the theoretical spectra of API and cyclodextrin were determined, and, on the basis of the high consistency, the characteristic band was assigned to all analyzed compounds. Spectra of all samples in the dataset are presented in the Appendix A. The spectra were cropped to range 400–2000 cm^−1^ and subjected to analysis with the machine learning approach.

### 2.2. Molecular Modeling

A molecular modeling approach was used in order to estimate the thermodynamics of complexation of the investigated prodrug β-lactam analogs–CDs systems. The optimized geometries of APIs along with the molecular electrostatic potentials (MEPs) calculated were presented in Figure 1. The optimized geometries and electrostatic potential maps for all investigated cyclodextrins are collected in the Appendix A. Optimized geometries of prodrug β-lactam analogs and CDs were set as initial geometries for docking using molecular dynamics. Preparation of initial complexes conformations was done by placing the β-lactam analog structure in 3D space facing one of the molecular domains towards one of two sides of cyclodextrins, resulting in a diverse set of conformations. The molecular domains considered as possible for inclusion in the docking study were identified in regard to the nonpolar (green) regions visualized on the MEPs, for CA: 1-acetoxyethyl and furyl groups, for CT: 2,2-dimethylpropanoyloxymethyl and methoxyiminoacetyl groups, and for PA: 2,2-dimethylpropanoyloxymethyl and phenol groups. The polar regions included for CA: [(Aminocarbonyl)oxy]methyl group, for CT: 2-amino-1,3-thiazol-4-yl group, and they were considered as possible domains susceptible to the association on the surface of CDs. The geometries of the conformations were optimized, revealing possible stable conformations according to the heat of formation calculated.

Heats of formation were calculated for each ingredient of complexes as well as for the complexes. The results of the final optimization step were used to calculate changes in standard enthalpy of complex creation according to the equation:$$\Delta_{r}H^{⊖}=\Delta_{tw}H_{API\cdot CD}^{⊖}-\left(\Delta_{tw}H_{API}^{⊖}+\Delta_{tw}H_{CD}^{⊖}\right)$$
where: $\Delta_{r}H^{⊖}$ is the standard enthalpy of the complexation reaction, $\Delta_{tw}H_{API\Delta CD}^{⊖}$ is the standard enthalpy of the complex creation, $\Delta_{tw}H_{API}^{⊖}$ is the standard enthalpy of the API creation, and $\Delta_{tw}H_{CD}^{⊖}$ is the standard enthalpy of cycoldextrin creation. The values were summarized in Table 2. Molecular modeling studies revealed binding modes of β-lactam analogs and CDs in complexes assuming a water environment. Binding modes of the investigated system were presented in the Appendix A. Significantly positive complexation energy indicating that complexation is unlikely was observed for CA–HPαCD. For all other cases, the results suggest that in a simulation environment of the isolated system, the complexation would occur.

### 2.3. Machine Learning

DSC is the method of choice for investigating cyclodextrin complexes as it provides reliable information about the complex formation in the sample by a simple comparison of several major thermal peaks on thermograms of the complexed system in a physical mixture. On the other hand, an application of a more specific spectral method such as tFTIR/ATR is limited due to complex, interpretively difficult dependencies between changes in spectra of isolated constituents, physical mixtures, and complexed samples. The motivation for the use of a machine learning technique lies in the application of tFTIR/ATR spectra as a specific fingerprint of the sample, carrying information about bands involved in complex formation. The presented algorithm utilized a dataset of known data constituted by spectra of samples divided to one of two classes—complexed and noncomplexed—depending on whether the DSC confirmed complex formation in the sample or not. Automated protocol of search for the most useful spectral bands ensured that only the significant vibrations would be taken into consideration for machine learning. The accuracy of the classification model in one-out testing was 90.1%, and the exact results are presented in Table 3.

Analysis of the importance of bands being used as input data for the classification model allowed the specification of spectral regions, which, according to the acquired model, are responsible for changes caused by the formation of the complex. The bands selected by the algorithm (Table 4) were matched with the bands on the FTIR/ATR spectra of pure β-lactam analogs, supported by theoretical calculations (Figure 2). The binding modes involving β-lactam analog domains indicated by the algorithm were simulated and compared against the most optimal ones acquired by docking. The binding enthalpies in relation to enthalpies of energetically favored binding modes in a water environment were reported in Table 2. Structures of binding modes derived from machine learning results are collected in the Appendix A.

Analysis showed that CA binds mostly with the axetil group, where interactions of carbonyl groups in the (acetyloxy)ethyl group and ((aminocarbonyl)oxy)methyl group are observed. The different mode of binding is possibly observed for the CA–HPβCD system. Molecular modeling studies showed that among all CA–CDs confirmed complexes, only CA–HPβCD showed positive enthalpy of binding when docked with the axetil domain included into the HPβCD lipophilic cavity. The machine learning model predicted a false negative for this example, which may suggest that the considered system has a unique binding mode in the whole dataset and cannot be predicted based on other examples. The most probable binding mode for CT according to machine learning analysis involves interactions of the pivoxil group and some interaction among the thiazole ring and cephem group with the cyclodextrins. The interactions of thiazole and cephem groups may be observed in Appendix A. The considered groups interact with polar, hydroxypropyl-rich moieties placed on the outer side of cyclodextrin. The algorithm predicted one false negative and one false positive for the CT systems. A false positive for a physical mixture of CT–HPγCD may indicate that spontaneous complexation occurred during storing, which is observed in systems of large, lipophilic CD such as HPγCD. The false negative for a system of mβCD–CT may indicate that a different binding mode than in all other samples is present in the experimental sample. PA complexed and noncomplexed systems were predicted entirely correctly. According to the analysis of the most important bands involved in the prediction of complexation, the pivoxyl group is mainly involved in the creation of the complex. The contribution of bands related to the C-H group in the β-lactam ring is also a significant variable for the classification model. Molecular modeling showed that deep inclusion may also involve a β-lactam ring in complexes with synthetic derivatives such as hydroxypropyl-substituted CD (Appendix A).

### 2.4. Discussion

The presented study demonstrates the potential application of machine learning algorithm in the area of CD inclusion complexes by elucidating binding modes of CA, CT, and PA in complexes with CDs. The identified molecular domains of CA indicate the axetil group as the main interacting domain for all CDs systems except HPβCD. According to results published by Shah et al., the axetil group may actually be involved in the interaction with macromolecules after complexation with HPβCD [30]. The reason for the divergence in these results may be due to differences in the preparation method and underlines the necessity of human supervision over computer-aided analysis. An approach for the fully computer-based elucidation of the binding mode of the CA–HPβCD complex was investigated in the study by Sapte et al [31]. The binding modes were simulated using the force-field-based method MMFF94, revealing that the most energetically favored mode involves the inclusion of CA’s furan ring. Application of the force-field method is a rough estimate of the outcome of a physical process of cyclodextrin complex preparation. The source of divergence may lay in the complexity of the inclusion phenomena, which is dependent on various environmental conditions unfeasible to estimate on the level of accuracy of static force-field simulation. The combined outcomes of discussed and presented papers allow us to draw the conclusion that theoretical simulation of the isolated system in certain binding modes does not cover conformational diversity related to the thermodynamics of complex inclusion in the real sample and is greatly affected by preparation method and intermolecular interactions. The simulation approach, to meet the requirements of a useful analytical technique, must be supported by an analysis of experimental results, whether it be conducted by human labor or by a machine learning algorithm.

## 3. Materials and Methods

### 3.1. Materials

Active pharmaceutical ingredients—cefuroxime axetil (CA), cefetamet pivoxil (CT), and pivampicillin (PA)—were synthesized in the Institute of Biotechnology and Antibiotics. Cyclodextrins, including αCD, βCD, γCD, HPαCD, HPβCD, HPγCD, and mβCD, were purchased from Sigma Aldrich Chemie. Complexes of API and cyclodextrins (1:1) were prepared by coprecipitation using saturated methanol solutions of APIs and saturated water solutions of CDs. The solutions of each binary system were shaken at 60 RPM in a 4.0 ml flask till complete evaporation of solvents and kept in a desiccator under a controlled humidity environment. 

### 3.2. Instrumentation

As a base method for the identification of prodrug β-lactam analogs–CDs systems, DSC was used. The thermal experiment was carried out with DSC apparatus TA Instruments DSC Q20, with the heating speed at 10 °C/min in the range 30–180 °C. The tFTIR spectra of complexes as well as pure substances constituting the given complex, along with their physical mixtures, were recorded with FTIR Bruker IFS 66v/S with a DTGS detector (Bruker, Billerica, MA, USA). The samples were put in in KBr pills in 1:100 ratio by applying 8 metric tonnes of pressure in a hydraulic press. The ATR FTIR spectra were obtained with ATR BRUKER VERTEX 70 (Bruker, Billerica, MA, USA) apparatus with DLaTGS detector directly on powder. The vibrational infrared spectra were recorded in the range between 400 and 4000 cm^−1^. 

### 3.3. Computation

The theoretical investigation consisted of several stages which allowed the acquisition of the optimal geometry of isolated molecules and the ingredients of the complexes. Initial conformation of the α-, β-, and γ-cyclodextrins were acquired from their crystal structure [32,33,34]. Initial conformations of the remaining cyclodextrins, as well as the APIs, were obtained with the application of a conformational search using a genetic algorithm implemented in Open Babel library [35]. Conformation search was followed by DFT B3LYP 6-311G(3df,3dp) geometry optimization of the most promising conformers of prodrug β-lactam analogs and DFT B3LYP 6-31G(d,p) for CDs using the Gaussian09 package [36]. Molecular docking and the relaxation part of the docked system through molecular dynamics were applied in order to establish stable conformations of complexes. The prodrug β-lactam analogs were divided into three subdomains branching out from penam or cephem substructures. The domains were placed facing two possible sides of cyclodextrins and optimized using MMFF94. Relaxation of manually docked structures was done in ChemBio3D 13.0 using molecular dynamics simulation lasting 100 ps with 2 fs timestep in a water environment to determine the most stable conformation. The dielectric constant of water 80.2 was used during the simulation. The length of the simulation was determined by an observation of two scenarios: for the unstable pose, a drift of the guest molecule away from CD, or for the stable pose, a drop in the root-mean-square deviation of distances between geometric centroids of the guest molecule and CD below 0.75 Å. For all simulated systems, the scenarios happened within 100 ps of simulation. The most energetically favored conformations of systems using MMFF94 were optimized with parametrized method 7 (PM7) implemented in MOPAC [37]. As a result, thermodynamically stable conformations were acquired and set as the geometry for calculation of FTIR spectra using PM7 as well.

An automated machine learning approach was applied for the selection of a classification model, allowing the optimizations of hyperparameters in regard to an accuracy metric. The goal of this approach was to maximize the accuracy of a classification of tFTIR/ATR spectra of given samples to one of two classes indicating that the sample contains cyclodextrin complex or is a physical mixture. The code in Python for data manipulation and processing was developed and is available upon request. The machine learning code was based on TPOT [38] and scikit-learn [39] libraries. The TPOT allowed us to select the best-suited algorithm from the ones available in scikit-learn and to optimize parameters of the model implementing the selected algorithm. The algorithm selected by TPOT was an ensemble of randomized decision trees, as implemented in the ExtraTreesClassifier class of scikit-learn. The default parameters of the model were optimal. Input data were multispectral tFTIR/ATR images in the range 400–2000 cm^−1^ of all samples, including complexed systems as well as physical mixtures for a total of 42 samples. The model was tested with a leave-one-out, cross-validation (LOOCV) approach. The LOOCV involved iterative testing of the model, where during each iteration, one of the samples was used for testing and the rest for training. The protocol involves an iteration through all of the 42 samples in a dataset dividing them into 41 training samples in each iteration and a test sample indicated by an iterator. The protocol used each sample as a test sample once throughout the whole testing procedure. The test score is then computed as a mean value from partial scores, and the model is randomly initialized on the beginning of each test iteration, preventing any data leaks from a training set to a test set.

## 4. Conclusions

Cyclodextrin complexes were successfully prepared using the coprecipitation method in 20 of 21 systems of prodrug β-lactam analogs. The DSC as the method of choice for the confirmation of complex identification was used. Both melting point temperature shifts, as well as changes in the intensity of endothermic peaks on the DSC curves, were considered as indicators for complex formation. The only system which did not expose any significant changes in the DSC curve of the complexed sample in relation to the physical mixture was CA–HPαCD; thus, it was considered as noncomplexed. tFTIR and ATR spectra were subjected to automatic analysis for the detection of an inclusion complex using a machine learning approach. The algorithm built according to the automatic machine learning paradigm was able to successfully predict cyclodextrin complex formation using only spectral data as inputs. Although learning of the model required DSC results as calibration data, only spectral results were used during testing, thereby calculating predictions reported in this study. Analysis of the importance of a given band for a trained classifier allowed us to explain the modes of binding which are observed on average for all investigated complexes of the given prodrug β-lactam analogs. Application of automated machine learning allowed us to achieve notably accurate results of more than 90% correctly classified spectra of prodrug β-lactam analogs–CDs systems to complexed/noncomplexed classes. The molecular modeling allowed us to confirm conclusions deducted from the machine-learning algorithm outcome. However, the most energetically favored binding modes predicted for isolated docked and optimized poses were not entirely in agreement with results acquired in the experiment. That suggests that environmental thermodynamic effects play a crucial role in the formation of particular binding modes in the experimental condition. The functionalities of computer algorithms based on the machine learning approach enable a better insight from the multispectral data gathered in experiments. A possibility to switch from a destructive thermal into a nondestructive, information-rich spectral method without additional human labor is a major milestone for the analysis of β-lactam analogs–CDs systems. 

## Figures and Tables

**Figure 1 molecules-24-00743-f001:**
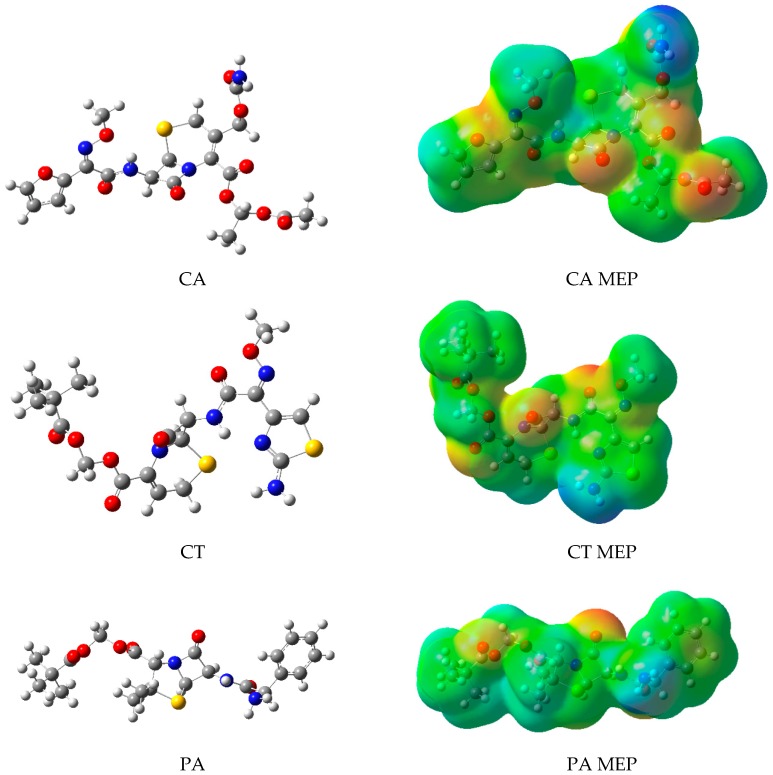
Optimized structure of prodrug β-lactam analogs and their maps of electrostatic potentials (MEPs).

**Figure 2 molecules-24-00743-f002:**
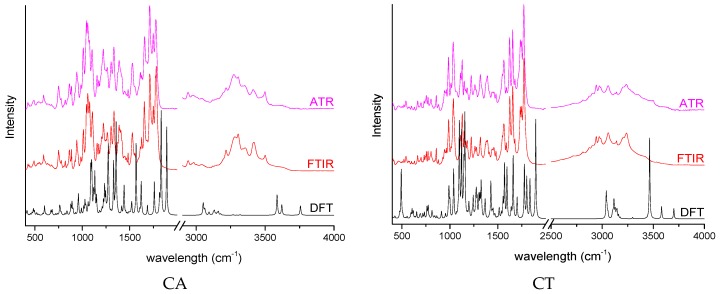
Experimental and theoretical spectra of CA, CT, and PA.

**Table 1 molecules-24-00743-t001:** Observed changes on the differential scanning calorimetry (DSC) thermograms of complexed samples, pure constituents, and physical mixtures.

	CA			CT			PA		
Cyclodextrin	Physical Mixture	Complex	Change	Physical Mixture	Complex	Change	Physical Mixture	Complex	Change
**αCD**	77 °C 138 °C	- -	Intensity ↓ Intensity ↓	71 °C 126 °C 164 °C	- - -	Intensity ↓ Intensity ↓ Intensity ↓	76 °C 131 °C	- -	Intensity ↓ Intensity ↓
**βCD**	114 °C	116 °C	Peak ↔ Intensity ↓	113 °C 164 °C	- -	Intensity ↓ Intensity ↓	120 °C	-	Intensity ↓
**γCD**	100 °C	103 °C	Peak ↔ Intensity ↓	86 °C 164 °C	100 °C -	Peak ↔ Intensity ↑ Intensity ↓	114 °C	100 °C	Peak ↔ Intensity ↓
**HPαCD**	-	-	No changes	76 °C 164 °C	93 °C -	Peak ↔, Intensity ↑ Intensity ↓	84 °C 138 °C	84 °C 140 °C	Intensity ↑ Peak ↔ Intensity ↓
**HPβCD**	83 °C	87 °C	Peak ↔ Intensity ↓	83 °C 164 °C	91 °C 164 °C	Intensity ↓ Intensity ↓	84 °C 138 °C	100 °C -	Peak ↔ Intensity ↑ Intensity ↓
**HPγCD**	91 °C	85 °C	Peak ↔ Intensity ↓	85 °C 164 °C	95 °C 164 °C	Intensity ↑ Intensity ↓	84 °C 138 °C	94 °C 145 °C	Peak ↔ Intensity ↓ Peak ↔ Intensity ↓
**MβCD**	89 °C	92 °C	Peak ↔ Intensity ↓	93 °C 165 °C	93 °C 165 °C	Intensity ↑ Intensity ↓	82 °C 138 °C	92 °C 145 °C	Peak ↔ Intensity ↑ Peak ↔ Intensity ↓

CA: cefuroxime axetil, CT: cefetamet pivoxil, PA: pivamipicillin.

**Table 2 molecules-24-00743-t002:** Binding enthalpies of complexes conformations acquired according to machine learning results (experimentally favored) in relation to conformations favored in a simulated water environment.

	Cefuroxime axetil		Cefetamet pivoxil		Pivamipicillin	
	Simulation Favored	Experimentally Favored	Simulation Favored	Experimentally Favored	Simulation Favored	Experimentally Favored
**α**	−157.31	−134.83	−239.54	−235.75	−197.79	−149.85
**β**	−238.22	−145.17	−279.69	−131.33	−263.39	−195.71
**γ**	−255.57	−196.71	−291.79	−178.73	−243.19	−125.94
**HPα**	486.92	558.60	−250.64	−182.03	−235.03	−180.66
**Mβ**	−256.15	−168.48	−185.90	−144.22	−236.37	−173.36
**HPβ**	−904.60	352.80	−268.65	−259.74	−193.11	−223.94
**HPγ**	−352.92	−230.65	−282.78	−263.54	−246.71	−293.35

**Table 3 molecules-24-00743-t003:** Testing predictions of the model, where True–complexed sample, False–physical mixture/noncomplexed sample.

	CA				CT				PA			
	Complexed		Physical mixtures		Complexed		Physical mixtures		Complexed		Physical mixtures	
	Predicted	True value	Predicted	True value	Predicted	True value	Predicted	True value	Predicted	True value	Predicted	True value
**αCD**	True	True	True	False	True	True	False	False	True	True	False	False
**βCD**	True	True	False	False	True	True	False	False	True	True	False	False
**γCD**	True	True	False	False	True	True	False	False	True	True	False	False
**HPαCD**	False	False	False	False	True	True	False	False	True	True	False	False
**HPβCD**	False	True	False	False	True	True	False	False	True	True	False	False
**HPγCD**	True	True	False	False	True	True	True	False	True	True	False	False
**MβCD**	True	True	False	False	False	True	False	False	True	True	False	False

**Table 4 molecules-24-00743-t004:** The spectral bands contributing to model, most important features first.

API	Peak [cm^−1^]	Theoretical	Description
**CA**	1772	1832	C=O *s* in (acetyloxy)ethyl group C=O *s* in ((aminocarbonyl) oxy)methyl group
	1264	1274	C–O *s* in (acetyloxy)ethyl group C–H *w* in ((aminocarbonyl) oxy)methyl group
**CT**	1774	1838	C=O *s* in pivoxil group
	1458	1479	CH2 *s*c in pivoxil group
	1277	1275	C-C *s* between the thizaol ring and cephem group C-N-H *b-ip* between the thizaol ring and cephem group
**PA**	1774	1838	C=O *s* in pivoxil group
	1458	1479	CH2 *sc* in pivoxil group
	1371	1362	C-H *b* in β-lactam ring
	1283	1293	C-C *s* in pivoxil group

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
