# Peer review of "Machine Learning Approach for Determining the Formation of β-Lactam Antibiotic Complexes with Cyclodextrins Using Multispectral Analysis"

_molecules, 2019, doi:10.3390/molecules24040743_

Round 1

Reviewer 1 Report

The begining of Introduction should be reconstructed to address the topic of the article.

What software and methodology was used for molecular docking and molecular dynamics. Did the MD last only 100 ps (or maybe 100 ns)?

What was the rationale of calculating MEP? It is not described in the manuscript.

Reviewer 2 Report

The authors present a method to classify tFTIR/ATR spectra according to whether they corresponding to a cyclodextrin-antibiotic complex or not. The work is in principle interesting but lacks essential information about computational procedures. It is essential that the authors provide the additional information listed below. The text is otherwise readable, though it features a couple of typos and clumsy sentences that should be amended (see minor comments). My background is on molecular modelling, so I will only comment on this aspect of the work.

Major comments:

1)     Machine learning:

a.      An “automated machine learning algorithm” is repeatedly mentioned in the text, but its name is never provided. Which is it? Neural Networks? Support Vector Machine? Random Forests? Something else?

b.      For whichever machine learning algorithm used, which parameters were adopted? Were any of those parameters profiled for better performance? If so, how?

c.      The scikit-learn package is mentioned, so I assume in house code was developed in Python. I suggest this should be indicated.

2)     Molecular modelling:

a.     “conformation search” is mentioned in methods, but no satisfactory explanation on how this was done is provided. From the main text, my understanding is that this was done manually. Is that the case? How were poses selected? How many of them? How many turned out to be stable?

b.      How was the examples dataset split into training and testing set? How many examples were featured in training and testing sets?

c.      If poses were manually assembled, it’s not because they were then relaxed with DFT and MD that they would become plausible. Despite that, it appears that the training set generated with this procedure was enough to lead to reasonably good performances from the machine learning algorithms. I suggest this point should be mentioned in the text. Is there any cyclodextrin-antibiotic complex of known structure, so that its simulated spectrum could be compared against the poses generated by modelling?

d.      Line 255: The expression “molecular docking using molecular dynamics” gives the false impression that the molecules were left to freely diffuse in solution until eventually binding to each other. This is however not the case, as MD was only used to relax previously generated arrangements. Please rephrase.

3)     Molecular Dynamics:

a.     Concerning the 100 ps molecular dynamics simulations, virtually no information is provided on the protocol. Please report at least the following. Which thermodynamic ensemble was simulated (NPT, NVT, NVE)? Using which thermostat/barostat? What was the simulation timestep?

b.      Was simulation convergence anyhow assessed? i.e., why 100 ps?

c.      A “water environment” for MD simulations is mentioned. Was a dielectric used, or the system was solvated with explicit water molecules? If the latter, how?

d.      Line 261: MMFF94 is not a “level of theory”, it is a force field.

Minor comments:

·        Line 38: this opening sentence is slightly confusing, I suggest starting with a more generic “Prodrugs are poorly on non-active chemical compounds which are biotransformed in vivo to […]”

·        Lines 39 and 44: please write the definition of acronym in the text, and the acronym itself within parentheses, and not the inverse.

·        Line 60: inside the molecule

·        Line 61: influence the physicochemical environment

·        Line 71: define here the acronym “DSC”

·        Line 71: FT-IR. In the rest of the text “FTIR” is usually used (besides in the abstract). Please pick one of the two.

·        Line 76: the major drawback of thermal techniques is their destructive nature

·        Lines 76-79: it feels these two sentences repeat each other.

·        Line 94: please define all the acronyms in this line (PCA, kNN, HCA).

·        Line 109: In this paper

·        Line 133: “Using the theoretical approach”. Which one?

·        Line 149: “anaogs” (typo)

·        Line 157: “envrinoment” (typo)

·        Line 173: “siginificant” (typo)

Round 2

Reviewer 1 Report

The manuscript can be accepted in the present form.

Author Response

Dear editor,

thank you for your positive feedback about the Manuscript. 

Best regards,

Judyta Cielecka-Piontek

Reviewer 2 Report

The authors have addressed most of my questions, and the manuscript has improved as a result. The following points should still be carefully considered.

1)     I must insist in asking the authors to explicitly address in the text the following points, raised in my first review:

a.       please state explicitly (perhaps in line 380) how many examples were used for training of the classifier.

b.      Point 3b asked: “Was simulation convergence anyhow assessed? i.e., why 100 ps?”. The response was: “The 100 ps was used as reasonable number of steps to finish equilibration of the system.” I am afraid this does not answer the question, unless an explanation of what makes 100 ps reasonable is provided. A way to answer this could be to either mention that convergence of some measurable quantities (e.g. RMSD, RMSF, dihedral angles, …) was observed or, at least, at least state that the molecule is expected to only undergo dynamics in the ps timescale.

2)     On line 358, the authors define their Python code as “proprietary”. This means that the code is not for distribution/its usage is restricted. If this is what the authors intended, why? This wouldn’t appear like good academic practice (i.e. at least make the software available upon request).

3)     Text amendments in this new version of the manuscript are ridden with orthographic and grammatical mistakes. I hope I will not sound excessively grumpy, but I should point out that simply using Microsoft Word’s autocorrect tool (or a free online tool like Grammarly) would have prevented all of them. I tried to list them here:

·        Lines 39-40: “AN Innovative approach FOR THE analysis cyclodextrin systems may be based on theoretical approach involving machine learning and molecular modeling. The theoretical approach to the analysis of the cyclodeXstrin systems is particularly valuable […]”. Besides the amendments indicated in bold, note that everything in italic is a repetition.

·        Line 42: “THE cavity of A cyclodeXstrin carrier”

·        Line 207: rEGArds

·        line 221: ingrEdient

·        line 335: aPParatus

·        line 345: two errors in one single edit: “RELAXATION part of DOCKED system […]”

·        line 348-349: “Relaxation of manualLy docked structures was done […]”

·        line 358: proPRIEary

·        line 376: devEloped

·        lines 358-378: this sentence makes no grammatical sense, I suggest splitting it in two. I note that this sentence is particularly important as it explains how the learning algorithm was selected.

·        line 379: impLemented

Author Response

Point 1. I must insist in asking the authors to explicitly address in the text the following points, raised in my first review:

a.       please state explicitly (perhaps in line 380) how many examples were used for training of the classifier.

Response 1a. The explanation on leave one out method was added (lines 294-301).

b.      Point 3b asked: “Was simulation convergence anyhow assessed? i.e., why 100 ps?”. The response was: “The 100 ps was used as reasonable number of steps to finish equilibration of the system.” I am afraid this does not answer the question, unless an explanation of what makes 100 ps reasonable is provided. A way to answer this could be to either mention that convergence of some measurable quantities (e.g. RMSD, RMSF, dihedral angles, …) was observed or, at least, at least state that the molecule is expected to only undergo dynamics in the ps timescale.

Response 1b. The rationale behind 100 ps simulation time was described in the manuscript (lines 276-280).

Pont 2. On line 358, the authors define their Python code as “proprietary”. This means that the code is not for distribution/its usage is restricted. If this is what the authors intended, why? This wouldn’t appear like good academic practice (i.e. at least make the software available upon request).

Response 2. The unfortunate word “proprietary” was removed, we are happy to share the code with other researchers.

Point 3. Text amendments in this new version of the manuscript are ridden with orthographic and grammatical mistakes. I hope I will not sound excessively grumpy, but I should point out that simply using Microsoft Word’s autocorrect tool (or a free online tool like Grammarly) would have prevented all of them. I tried to list them here:

Response 3. We are sorry for editing errors and typos in the manuscript, we put too much faith in autocorrecting tools, which turned out not to work in review mode of our text editor. Nevertheless, the manuscript was once again checked for errors and we have made every effort to correct them all.

·        Lines 39-40: “AN Innovative approach FOR THE analysis cyclodextrin systems may be based on theoretical approach involving machine learning and molecular modeling. The theoretical approach to the analysis of the cyclodeXstrin systems is particularly valuable […]”. Besides the amendments indicated in bold, note that everything in italic is a repetition.

The sentences were corrected.

·        Line 42: “THE cavity of A cyclodeXstrin carrier”

The words were corrected.

·        Line 207: rEGArds

The word was corrected.

·        line 221: ingrEdient

The word was corrected.

·        line 335: aPParatus

The word was corrected.

·        line 345: two errors in one single edit: “RELAXATION part of DOCKED system […]”

The words were corrected.

·        line 348-349: “Relaxation of manualLy docked structures was done […]”

The word was corrected.

·        line 358: proPRIEary

The word was corrected.

·        line 376: devEloped

The word was corrected.

·        lines 358-378: this sentence makes no grammatical sense, I suggest splitting it in two. I note that this sentence is particularly important as it explains how the learning algorithm was selected.

The sentence was rewritten in a more clear way.

·        line 379: impLemented

The word was corrected.

Best regards,

Judyta Piontek